# West London Healthy Home and Environment (WellHome) Study: Protocol for a Community-Based Study Investigating Exposures Across the Indoor-Outdoor Air Pollution Continuum in Urban Communities

**DOI:** 10.3390/ijerph22020249

**Published:** 2025-02-10

**Authors:** Diana Varaden, Benjamin Barratt, Margaret J. Dallman, Adam Skillern, Munira S. Elmi, David C. Green, Anja H. Tremper, Michael Hedges, William Hicks, Max Priestman, Leon P. Barron, Shane P. Fitzgerald, Holly M. Walder, Stephanie L. Wright, Ian S. Mudway, Matthew C. Fisher, Samuel J. Hemmings, Wouter Poortinga, Francesca Tirotto, Sean Beevers, Heather Walton, Tuan Vu, Klea Katsouyanni, Dimitris Evangelopoulos, George Young, Dylan Wood, Christopher Griffiths, Frank J. Kelly

**Affiliations:** 1Environmental Research Group, School of Public Health, Imperial College London, London W12 0BZ, UK; b.barratt@imperial.ac.uk (B.B.); a.skillern21@imperial.ac.uk (A.S.); m.elmi21@imperial.ac.uk (M.S.E.); d.green@imperial.ac.uk (D.C.G.); anja.tremper@imperial.ac.uk (A.H.T.); m.hedges20@imperial.ac.uk (M.H.); will@airawarelabs.com (W.H.); m.priestman@imperial.ac.uk (M.P.); leon.barron@imperial.ac.uk (L.P.B.); s.fitzgerald@imperial.ac.uk (S.P.F.); h.walder21@imperial.ac.uk (H.M.W.); s.wright19@imperial.ac.uk (S.L.W.); i.mudway@imperial.ac.uk (I.S.M.); s.beevers@imperial.ac.uk (S.B.); h.walton@imperial.ac.uk (H.W.); tuan.vu@imperial.ac.uk (T.V.); k.katsouyanni@imperial.ac.uk (K.K.); d.evangelopoulos@imperial.ac.uk (D.E.); g.young24@imperial.ac.uk (G.Y.); dylan.wood@imperial.ac.uk (D.W.); frank.kelly@imperial.ac.uk (F.J.K.); 2MRC Centre for Environment and Health, Imperial College London, London W12 0BZ, UK; 3NIHR HPRU in Environmental Exposures and Health, Imperial College London, London W12 0BZ, UK; 4Department of Life Sciences, Imperial College London, London SW7 2AZ, UK; m.dallman@imperial.ac.uk; 5NIHR HPRU in Chemical and Radiation Threats and Hazards, Imperial College London, London W12 0BZ, UK; 6MRC Centre for Global Infectious Disease Analysis, Imperial College London, London W12 0BZ, UK; matthew.fisher@imperial.ac.uk (M.C.F.); s.hemmings@imperial.ac.uk (S.J.H.); 7Centre for Climate Change and Social Transformations, School of Psychology, Cardiff University, Cardiff CF10 3AT, UK; poortingaw@cardiff.ac.uk (W.P.); tirottof@cardiff.ac.uk (F.T.); 8Department of Hygiene, Epidemiology and Medical Statistics, School of Medicine, National and Kapodistrian University of Athens, 115 28 Athens, Greece; 9Asthma UK Centre for Applied Research, Barts Institute of Population Health Sciences, Queen Mary University of London, London E1 4NS, UK; c.j.griffiths@qmul.ac.uk; 10MRC and Asthma UK Centre in Allergic Mechanisms of Asthma, King’s College London, London WC2R 2LS, UK

**Keywords:** indoor air pollution, exposure modelling, asthma, respiratory symptoms, behaviour change, immunotoxicity, participatory research

## Abstract

The relationship between indoor air quality and public health remains under-researched. WellHome is a transdisciplinary community-based study that will engage with residents to co-design feasible and acceptable research to quantify air pollution exposure in 100 homes in West London and examine its potential to exacerbate asthma symptoms in children. Sampling strategies such as using air quality monitors and passive samplers placed in kitchens, children’s bedrooms, and living rooms, will be developed in collaboration with local ambassadors and participating households to measure multiple physical, chemical, microplastic, and biological contaminants. This will provide a comprehensive understanding of indoor air quality across the city’s socio-economic gradient. Other data collected will include housing types and tenure, ventilation practices, occupant behaviours, time-activity, and airway symptoms. Epidemiological analysis will examine air pollution exposure impacts on children’s respiratory health. The particulate mixture’s relative hazard will be evaluated in toxicity studies based on source profiles and activity patterns of participants, focusing on asthma exacerbation related pathways. The study’s findings will be communicated to participants through co-designed reports and inform evidence-based recommendations for reducing indoor air pollution in London and urban areas worldwide. By raising awareness and providing actionable insights, WellHome seeks to contribute to global efforts to improve the health and well-being of vulnerable communities.

## 1. Introduction

### 1.1. Background and Rationale

Indoor air pollution can significantly contribute to overall air pollution exposure, especially for individuals who spend a significant amount of time indoors [1]. This is particularly concerning for people with asthma, as exposure to diverse indoor pollutants, from allergens to volatile organic compounds (VOCs), can trigger or exacerbate asthma symptoms [2,3]. These pollutants can lead to airway inflammation, increased mucus production, and make breathing difficult [4]. Furthermore, contaminants can interact, amplifying their negative health effects on individuals with asthma [5,6,7]. Despite this impact, especially among vulnerable groups, such as young children, pregnant woman, individuals with lung conditions, the elderly, and those from disadvantaged communities [1,8], it remains a relatively under-researched area, with historically a greater emphasis placed on outdoor air quality. Moving from outdoor to indoor environments presents significant challenges to public health research [1]. An ecological problem becomes a more personal one, in that homes are private spaces where people exercise their behaviours and daily routines, such as smoking, cooking, heating, and cleaning. Policies encroaching upon that choice can be unpopular and inappropriate for households with specific circumstances [9]. Socio-economic circumstances can play a significant role in how much choice households have regarding neighbourhood, housing quality, insulation, home technology, etc., leading to disparities in exposure [1,10]. These challenges need to be considered if we are to understand more fully the potential health impacts of indoor air pollutants on the population, especially within vulnerable groups, and to inform the design of effective measures to make home environments healthier.

There is an increased awareness that a holistic consideration of the impacts of air pollution on health requires an understanding of the continuum of exposures across the indoor and outdoor environment and life course [8]. This extends beyond the infiltration of ambient pollution into the home, school, or workplace to consideration of the indoor sources and their contribution to the highly heterogeneous indoor environment. Although the literature demonstrating the adverse effects of ambient pollution is extensive and mature, reflected by a joint European Respiratory Society/American Thoracic Society policy statement [11], work on indoor air pollution is less evolved [8]. Understanding the way indoor sources and infiltration efficiency affect exposure to both indoor and outdoor generated pollution and developing predictive models that can be extended to several types of households may be a way to move forward.

### 1.2. A Community-Driven Approach to Indoor Air Pollution Research

The limited research on indoor air pollution can at least partly be attributed to the challenges associated with conducting studies in personal spaces, such as people’s own homes, which require resident consent and participation. Homes are private areas governed by occupant behaviours and routines that are influenced by various social, physical, cultural, and economic factors. This individuality complicates research without imposing artificial constraints that may affect the outcomes. Researchers need resident consent to install monitoring equipment, but the intrusion into personal spaces often deters participation. Furthermore, public awareness of indoor air pollution is generally lower than that of outdoor air pollution. As a result, people may be less inclined to engage with the topic, as they might not see indoor air pollution as a problem.

To address these challenges, the West London Healthy Home and Environment Study (WellHome) has been co-designed with representatives of the local community from the outset. In 2020, we conducted a pilot study funded by a UKRI Citizen Science grant (“My house, my rules: Co-designing residential air pollution research”) [12] to ensure our study design was acceptable and feasible. By involving residents from the start, we developed a research methodology enriched by local knowledge. Residents highlighted that research on indoor air quality should: (a) answer the question of “*whether they should let the pollution in or let the pollution out*”, (b) prioritise participation of families with young children and marginalised communities, (c) include local ‘ambassadors’ as key members of the research team, and (d) ensure that study results are shared with participants. Guided by these insights and a community-based approach, the WellHome team will actively work with residents throughout the research project. This approach will foster collective ownership and offer opportunities for education and action on air pollution. At the same time, principles of good research practice will be observed in exploring associations between environmental exposures and health outcomes and the application of a health impact assessment.

The WellHome study consists of two core components: measurements in 100 homes, conducted over two 28-day periods, and more detailed monitoring in an additional 10 priority homes, where measurements will be taken continuously over 12 months. These 10 homes will be recruited separately from the initial cohort and will have their own inclusion criteria. These criteria include:At least one child in the household must have a diagnosis of asthma by a health practitioner.Participants should expect to reside at the same address for a minimum of 12 months.There must be no smokers in the household.Similar to the 100 homes group, participants must reside in West London.

The protocol outlined in the following sections focuses exclusively on the work planned for the measurements within the 100 homes.

### 1.3. Study Aim and Objectives

WellHome aims to investigate air pollution exposures in children and adults across the indoor-outdoor continuum within vulnerable and diverse urban communities in West London, with a focus on households with children affected by asthma and allergies. Based on this information, the health impacts of exposure to air pollution on children’s respiratory health, and the relative hazard of indoor/outdoor pollutant sources, we will develop mitigation measures aimed at reducing exposures, alleviating respiratory symptoms, and improving overall well-being. The WellHome project consists of six complementary work packages (WPs) (Figure 1), which collectively address the following objectives:to identify and explore contrasts in exposure to chemical and biological contaminants within heterogeneous households, considering indoor and outdoor measurements, household characteristics, area level deprivation, seasonality, proximity to traffic sources, and children’s time-activity to estimate personal exposures.to understand the potential of these exposures to affect the health of children with asthma and allergies in these households through epidemiological evaluation of symptoms and toxicological assessment of the relative hazard of chemical and biological aerosols.to develop behavioural and policy recommendations aimed at reducing the potential health burdens arising from indoor and ingressed outdoor air pollutants.

## 2. Materials and Methods

### 2.1. Study Setting and Participants Recruitment

We will engage with local communities within an ethnically diverse area of West London. This area was selected as it has high-density housing and pockets of significant deprivation and health inequalities. Engagement will take place in partnership with the Imperial College London White City Engagement Team (WCET), our local partners Nova [13] and Bubble and Squeak [14], and other community groups which are part of the WCET network (e.g., arts and crafts groups, knitting groups, the Big Local group, activist women’s groups, gardening groups, etc.). This will give us the opportunity to engage with a diverse range of community members. To support our community engagement and recruitment strategy, ten WellHome Ambassadors will be recruited from the local community. With the help of the WCET, our local partners, and the ambassadors, we will recruit a cohort of 100 households based on the following inclusion criteria:Households must reside in the area surrounding the Imperial College White City Campus in West London.Participants must be over 18 years old and be able to provide informed written consent.Participant must be the primary householder.We will prioritise the participation of households with children aged 5 to 17 years who have asthma and allergies, with a particular focus on families from minority ethnic backgrounds. This approach aims to ensure a diverse participant group that reflects the community’s diversity.

The main inclusion criterion for children:Participating children who have asthma or allergies must reside in the participating household.

We will provide potential participants with a Participant Information Sheet (PIS) and instructions on how to complete an expression of interest form if they wish to register to take part in the study. Expression of interest forms will be reviewed based on the criteria (as highlighted above) as well as ensuring we have a group of homes that broadly reflect West London’s rich cultural diversity and from a range of socio-economic backgrounds. Potential participants will be informed that it may not be possible to involve everyone who registers. Households that are selected to take part in the study will be provided with a consent form to sign and further details about their participation. Verbal consent would be obtained if requested by the participant to facilitate participation to those who may lack literacy skills. If verbal consent is attained, a consent form will be completed by a WellHome Team representative on behalf of the participant over the phone and in person. When possible, we will offer a translator to assist with the recruitment process. Children under the age of 18 living in the participating household will receive age-appropriate PIS(s) and will be asked to provide assents to participate. Participation in the study is voluntary, and participants will be able to stop at any point without the need to state a reason. Participants will be provided with compensation of £100 for their time. This approach to participant compensation has been approved by our ethics committee.

### 2.2. Participant Safety and Anonymity

All participants’ personal information will be securely stored and will not be disclosed to unauthorised individuals. We will use a locked filing cabinet in a locked office for paper-based personal data. Digital data will be password-protected. Data will be pseudonymised for processing and analysis purposes, with personally identifiable information and associated IDs stored securely and separately from the main dataset. Researchers visiting the households will always be accompanied by another team member and will never visit the home without the main householder present. Additionally, they will not be left alone with unattended minors at any time. If any concerns arise regarding the welfare of a child or vulnerable adult participating in the study, these will be reported to the local safeguarding officer in accordance with the College’s guidance and referral process. Ethical approval has been obtained from the Imperial College Research Ethics Committee (ICREC), Reference number 21IC7385, 22 June 2022.

### 2.3. Study Design

The study will follow a purposive sampling strategy to capture data from a broad spectrum of housing types and tenures. We will install 60 indoor air quality monitors in rotation across the 100 participating homes, each for two 28-day periods separated by three to six months, ensuring data is collected at two different time points in each home (Campaign 1 and Campaign 2). Householders will be invited to complete a series of questionnaires at different time points throughout the study. Some of these questionnaires will involve the active participation of children, typically those experiencing asthma and allergies symptoms (e.g., time-activity and daily symptoms diary). Figure 2 below shows a graphical representation of the study design.

The following sections outline the work that will be undertaken in each work package (WP). We plan to include a number of postgraduate students across the project, across the different work packages.

### 2.4. Engagement and Educational Activities (WP1)

To raise awareness and understanding of air pollution in the study area, this WP comprises various educational, engagement, and involvement activities and organised events that will be open to all local residents, regardless of whether they are participating in the WellHome project or not. These events will constitute part of our engagement programme and will be designed in collaboration with our local partners and ambassadors. The engagement programme will commence before recruitment begins and continue throughout the full duration of the study. Examples of the engagement activities that will be delivered are:STEM (Science, technology, engineering, and mathematics) themed activity packs and materials about our research that can be done at home, produced by WellHome in partnership with the Indoor Air Quality Working Party [15] and The Invention Rooms at Imperial College London [16].Face-to-face and online educational sessions for children.Workshops with children to learn about the nature of the data we are collecting and how these data will be analysed and interpreted, providing an opportunity for children to actively engage in scientific investigations, improving mathematical literacy, and nurturing their curiosity in science.Interactive panel discussions during which residents will have opportunities to meet with WellHome researchers to ask questions about the study.Ad-hoc workshops with participating families aimed at actively gathering their input into the development of the study’s public-facing reports.An online community air quality research hub, incorporating live data feeds, advice, participant profiles, and notifications.

### 2.5. Household Exposure Sampling (WP2)

This WP will utilise a comprehensive set of air quality monitors, passive samplers, and analytical techniques, alongside newly developed laboratory analysis methods, to quantify the exposure of the cohort to a range of chemical and biological asthma triggers, as well as other contaminants within their homes.

#### 2.5.1. Air Pollution Monitoring Network

PM_2.5_, NO_2_, HCHO, tVOCs, CO_2_, temperature, and humidity will be measured using indoor air quality monitors at 1-min time resolution in three locations inside each of the 100 homes cohort. Dyson air purifiers (Models DP04 and TP09, Dyson Ltd., Malmesbury, UK) modified to remove the air cleaning functionality, as shown in Figure 3a, will be used to measure PM_2.5_, NO_2_, HCHO, and tVOCs. These monitors are integral to the commercial purifier and designed to require no input from the participants; this approach was chosen to allow for easy integration into a future air purification study. Data collection will be automated via a WiFi hub installed in each home. CO_2_ will be measured independently (SCD4x, Sensirion, Stäfa, Switzerland).

On visit 1 and visit 3, the installation day for the two respective campaigns (see Figure 2), we will install three sets of monitors in each participating home: one set each in the kitchen, living room, and the bedroom where the child with asthma or allergies sleeps. Information about the participants’ homes, including a photograph of monitors in situ, will be taken during these installation visits. Additionally, the distance between the monitors and potential sources of pollution (e.g., external windows and doors, gas hobs, toaster, air fryer, kettle, rice cooker, microwave, and ovens) will be recorded. Monitors from each participating home will be collected at the end of each of the 28-day periods (visit 2 and visit 4).

All monitors utilised in the study will be benchmarked against reference or higher grade instrumentation, or independent laboratory analysis for PM_2.5_ (Partisol 2025, Thermo Fisher Scientific, Waltham, MA, USA); filter weighing using BS EN16450; Fidas (E200, Palas, GmbH, Karlsruhe, Germany), NO_2_ (N500, Teledyne Technologies Inc., Thousand Oaks, CA, USA), CO_2_ (LI-820, LI-COR Environmental UK, Cambridge, UK), tVOCs (ISO 16000-6:2021 [17], Enthalpy Analytical, Durham, NC, USA), and Formaldehyde (Enthalpy Analytical, NC, USA). This will be undertaken at the start and end of the programme, as well as approximately halfway through, and will utilise a bespoke calibration facility to ensure all monitors are both cross-compared and benchmarked together as efficiently as possible. Additional experiments will assess the response of the monitors to different sources, such as cooking, cleaning, and VOCs. Measurements will be compared to established indoor air quality standards (e.g., BS40102) [18].

In addition to the indoor air pollution measurements, an outdoor air quality monitoring network will be established, deploying Clarity Node-S monitors as part of the Breathe London Network [19]. These will be strategically placed in key outdoor locations within the surrounding community and spatial coverage will be optimised for the proposed core study area. Outside this core study area, additional monitors will be deployed close to each household. Each monitor will be equipped to measure PM_2.5_ and NO_2_.

#### 2.5.2. Passive Sampling Network

Novel, low-cost passive sampler arrays, as shown in Figure 3b,c, will be placed alongside the indoor air quality monitors for time-integrated capture of an exposure estimate over the 28-day period of thousands of chemicals, as well as biological and microplastic contaminants, both indoors and outdoors.

Passive sampling of chemicals is based on a diffusion-based technique that accumulates VOCs in the gas phase on selective sorbents over time (i.e., not via deposition like mould or microplastics particles). The accumulated fraction on these sorbents therefore represents a time-weighted average of [VOC] in the environment in which the device is deployed for the entire period. For the chemical passive samplers, these are based on a previous housing design used for water sampling which houses up to five separate sorbents in a disk format [20], but fitted with VOC-selective traps for applications in air [19]. Sorbents are manufactured by fitting the housings with 9 mm circular Emfab disks (punched out using a clean, dry wad punch) that are coated with a solution of 40 mg/mL Tenax in dichloromethane and allowed to dry. They are conditioned with methanol before deployment to remove contamination and wrapped in aluminium foil.

In this study, the passive sampler device houses five 9 mm diameter planar sorbent traps that will be deployed in the living room at each home [20]. Of the five sorbent traps, three will be liquid Tenax coated-Emfab disks for volatile and semi-volatile organics [21] and two will be silicone for volatile organics. The devices will be prepared and assembled immediately before deployment. The incorporation of two different sorbent types into the sampler allows a larger chemical range to be captured. These chemical passive samplers will be covered with aluminium foil until deployment, when they will be uncovered and attached to the Dyson air quality monitors. Each passive sampler device will have a unique code stamped into a blue cable tie used to attach it to the monitor (Figure 3c). Sample collection will start upon removal of the foil and will be completed at the end of the 28-day measurement period. Upon collection, the samplers will be re-covered, transported back to the laboratory that day, and frozen at −20 °C in the dark until analysis to preserve the stability of trapped chemical traces.

For biological sampling, deposition tray samplers (four sterile circular petri dishes of 140 mm diameter) will be used following a method adapted from our UK-wide Citizen Science Air-Quinox surveillance [22,23].

Sample collection of airborne microplastics is based on the deposition of microplastic particles onto glass fibre filters over a 28-day period [24]. After this period, the Pyrolysis—GC-MS analysis of the microplastics accumulated on the filter represent the total particle concentrations across the whole 28-day period. A time-weighted daily average of [microplastics] in the environment will subsequently be calculated by dividing this signal by the number of days in the sample collection period. This sampler collects naturally deposited particles, including microplastics, on a quartz microfibre filter (110 mm diameter) in a glass petri dish (120 mm diameter). The filters will be conditioned in a muffle furnace at 600 °C for one hour to remove surface impurities prior to deployment. The glass Petri dishes and their matching lids will be cleaned with general washing detergent and running tap water and will be soaked overnight in appropriately diluted alkaline laboratory detergent. The next day, they will be first rinsed 20 times with running tap water and then with distilled water before placing in a covered drying cabinet. The filters will be placed into the glass dishes in a laminar flow hood in a clean room. The operator will wear a cleanroom lab coat and shoes over cotton clothing. After the filters are placed, the glass lids will be placed on the Petri dishes. Sample labels will be placed on the lids and bottom of the Petri dishes. The samplers will then be wrapped in fresh aluminium foil and transported to the sampling site with the travel blanks. Each sampler type will have a field control blank held in storage. We will place one set of passive samplers (all three types) in each of the participating homes in the living area during each campaign (Figure 3b).

Where appropriate and where rain-free covered areas are available, a subset of homes will be asked to place these samples outdoors as well as indoors.

Once the sets of passive samplers have been collected, we will carry out chemical, bioaerosol, and microplastics analysis in the laboratory. Chemical analysis will be carried out via non-target screening using gas chromatography-mass spectrometry. Chemical compounds that will be characterised by this methodology will include hundreds of thousands of thermally stable and (semi-)volatile chemicals, including flame retardants, plasticisers, personal care products, household cleaning products, cooking, burning and heating-related chemicals, odours, and any substances arising from outdoor ingress. Microplastics will be characterised using pyrolysis-two-dimensional gas chromatography time of flight mass spectrometry. This method enables us to target twelve plastic polymers and detect their presence [25]. Bioaerosol burden will be assayed using primers for the 16S (Bacterial) and 18S rDNA (Fungal) to quantify absolute abundances using quantitative PCR [26,27].

#### 2.5.3. Questionnaires

All participants will be asked to complete a series of questionnaires at different time points throughout the study, as indicated in Figure 2. Data gathered from these questionnaires will allow us to contextualise each household’s exposure. To facilitate participation across different households with different needs or preferences, participants will be able to complete the questionnaires online (Qualtrics) or in paper form. Some of the questionnaires will also be translated into various local languages, such as Somali and Amharic, and where necessary we will recruit a translator and/or British Sign Language (BSL) interpreter. All general questionnaires are available in Appendix A. Below is a brief description of each one:

General household questionnaire 1 (visit 1), will gather specific information about each household, including their socio-economic and cultural background, and their behaviours (e.g., types of ventilation, windows, fuel used for cooking and heating, etc.).

General household questionnaire 2 (visit 2 and 4), will include questions about participants’ behaviour over the last 28 days (when measurements were taking place), including cooking habits, cleaning practices, opening windows, etc.

General household questionnaire 3 (visit 3), will ask participants if there were any substantial changes in their home between campaigns, such as a new family member, a new pet, or any home improvements.

The general questionnaires described above will be administered by the researcher during the indicated household visits (Figure 2).

Health Questionnaires, the International Study of Asthma and Allergies in Childhood (ISAAC) questionnaire will be administered to each child aged 5 to 17 years residing in the participating homes during visit 1. This questionnaire is available in two formats, a parent-completed version for children (which will be used in WellHome for children aged 5–11), and a self-completed version designed for young people aged 13–14 (which will be used in WellHome for young people aged 12–17). The questionnaire will enable us to assess the prevalence (‘ever had asthma’) and severity of asthma and allergies (attacks over the past 12 months) within this population [28]. If the child/young person is not present in the household at the time of the visit, a link to the questionnaire will be sent to the participant (via email or by text) so they can complete it with their children within the next 24 h.

The Asthma Control Test (ACT) will be administered to children aged 5–17 years residing in the participating home who have reported having asthma. This validated questionnaire will be conducted four times, at the beginning and end of each of the two campaigns. By assessing asthma control over the previous four weeks, the ACT will allow us to track changes in asthma control over time, and in relation to household exposure. For children aged 5–11, it includes 7 questions (4 answered by child; 3 by parent) and yields a maximum score of 27, which can be interpreted as follows: 0–12, very poor control; 13–20, poor control; >20, good control. For children aged 12–17, it includes 5 questions (answered by child) each scored out of 5. The maximum score is 25 and can be interpreted as follows: 0–15, very poor control; 16–20, poor control; >20, good control. The minimum clinically important difference is 2 points in children/adolescents up to 17 years [29].

The Paediatric Asthma Quality of Life Questionnaire (PAQLQ) will also be administered to children aged 5–17 years who have reported having asthma. This questionnaire will also be conducted four times, at the beginning and end of each of the two campaigns. The PAQLQ measures QOL in three domains (symptoms, activity limitation and emotional function). Children will be asked to think about how they have been during the previous week and to respond to each of the 23 questions on a 7-point scale (ranging from 7 = “not bothered at all” to 1 = “extremely bothered”). The overall PAQLQ score is the mean of all 23 responses and the individual domain scores are the means of the items in those domains. A change in score greater than 0.5 on the 7-point scale can be considered clinically important [30].

The ACT and PAQLQ must be completed with the child/young person present, therefore, if the child is not present in the household during the visit, a link to the questionnaire will be sent to the participant (via email or by text) so they can complete it with their children within the next 24 h.

The Test for Respiratory and Asthma Control (TRACK) will be administered to all children under 5 years old in the participating household, both at the beginning and at the end of each campaign. TRACK measures respiratory/asthma control in pre-school children. The first 3 questions relate to the last 4 weeks, the 4th relates to 3 months, and the 5th to one year. A score less than 80 means respiratory symptoms are not under control [31]. This questionnaire can be completed with the parent or guardian without the child being present.

Time-activity and daily symptoms diary, children aged 5 to 17 years living in participating households who have asthma or display allergic symptoms (allergic symptoms may include wheezing or whistling in the chest, a problem with sneezing, a runny or blocked nose when they DID NOT have a cold or the flu, or an itchy rash coming and going for at least six months) will be given an activity diary. In this diary, children will record their daily asthma symptoms, including the occurrence of cough, wheeze, shortness of breath, and nasal congestion. Additionally, they will document any symptoms of fever, the number of inhaler puffs used each day, and instances of school absenteeism. The diary will also capture general information about the children’s daily activities, including the location, duration, and nature of these activities, throughout the two 28-day sampling periods. The micro-environments included in the diary are classified as Home, Outdoors, In Transportation, and Other Indoors (not at home). The activity diary has been used successfully in a previous study [32] and is available in Appendix A. The objective of this diary is to allow a more nuanced interpretation of the sampling data, providing insights into modifiable behaviours involved in air pollution exposure indoors, but also to estimate children’s personal exposure to air pollution accounting for their mobility across the day. If there are no children with asthma or allergy symptoms, we will provide one activity diary to be completed by at least one child from the participating household. Older children (12 to 17 years) may prefer to complete the dairy by themselves. A £10 voucher from a stationary retailer will be offered to families as an incentive for completing the diary.

Ethnographic data will be collected during our visits to the households and includes the use of participant observation. This method involves the researcher visiting the homes closely observing the space, interactions, and conversations, and taking ethnographic fieldnotes [33]. Individuals’ identities will be kept confidential in the fieldnotes, and all later analysis and writing will use pseudonyms.

Perception questionnaire (visit 1, 2, 3 and 4), will gather insights into people’s views, perceptions, and worries about air pollution, their health and their environment (e.g., satisfaction with home and neighbourhood). The perception questionnaires will include questions designed to assess various components of the Health Belief Model in relation to health behaviours associated with indoor air pollution [34]. A long version of the questionnaire will be administered at the beginning of each monitoring period (visit 1 and 3), and a shorter one will be administered at the end of each monitoring period (visit 2 and 4). Data from the perception questionnaires will be used to identify psychological, social, and contextual predictors of behavioural changes during the study. Participants will complete the questionnaires online prior to each visit using a provided link.

### 2.6. Quantitative Profiling of Social Health Inequalities and Policy Disconnects Using Toxicological Paradigms (WP3)

This WP will assess the relative hazard of indoor and outdoor air pollution (ambient and experimentally generated source specific aerosols) and evaluate the impact of air pollution mitigation measures with the aim of developing new tools to inform future policy decisions. For this purpose, we will employ a multi-faceted approach, including: (1) enhanced chemical characterization—analysing particulate matter (PM) samples for metals, organics, biological components, oxidative stress, and microplastics; (2) in vitro toxicological screening—assessing the toxicity of PM samples using primary bronchial epithelial cells; and (3) in vivo toxicological evaluation—examining upper airway responses to PM exposure in individuals with asthma. Much of this work will be based on the 10 priority homes, which are not covered in this protocol paper, but the work will be informed by data collected on sources and time-activity patterns across the 100 homes. A more detailed description of the approaches taken to the hazard ranking of indoor sources and PM mixture are provided in Appendix A.

### 2.7. Occupant Understanding and Behavioural Factors in Indoor Air Quality (WP4)

Data from the perception questionnaires will be used to identify psychological, social, and contextual factors predicting behavioural changes observed during the WellHome study, as captured by the questionnaires in the second campaign. Specifically, mixed-effects models will be applied to examine how perceptions of indoor air quality, perceived vulnerability, severity of exposure to poor indoor air quality, and self-efficacy in improving indoor air quality evolve across the two campaigns of the project, and how these factors are associated with behavioural changes aimed at improving indoor air quality. This analysis will offer insights into the impact of engagement within the WellHome project on driving behavioural changes over time and will identify modifiable behaviours that can be influenced through such engagement, providing guidance for more effective future targeted intervention.

The results of this work package, together with insights from the engagement activities with the households, will inform the design of a nation-wide survey involving a sample of 1500 participants representative of the overall UK population, with an additional booster of 500 participants to increase the sizes of minority ethnic groups in the sample (Asian/Asian British, Black/African/Caribbean/Black British, White—Other, Mixed/Multiple ethnic groups, and other ethnic groups). The national survey will cover similar topics and questions that were used in the WellHome perceptions surveys. This aims to scale up the research findings to a national level, providing crucial data to support modelling efforts in predicting the impact of exposure reduction scenarios at the UK level (WP5).

### 2.8. Characterising Sources and Behaviour That Reduce Exposure in the West London Community (WP5)

This work package aims to develop an integrated indoor model to identify the main sources of indoor air pollutants and the primary exposure pathways to reduce exposure. Firstly, we will apply a mass balance model to the dataset of air pollutants (PM_2.5_, PM_10_ and NO_2_) compiled from the indoor measurements, outdoor measurements (sourced from Breath London and the London Air Quality Monitoring Network), and outdoor modelling data from our CMAQ-Urban model to evaluate the dynamics of indoor pollutants, including their indoor emission and decay rates for individual homes [35]. The data on indoor CO and CO_2_ measurements, along with indoor climate data (such as room temperature and humidity), will be used to assess human physical activities related to the use of indoor combustion sources, such as cooking and heating, as well as ventilation settings. We will then develop a multizone indoor air quality model based on the NIST-CONTAM model to predict the West London Community’s exposure to indoor PM and its compositions, and NO_2_ for year periods (2021–2022). Emission and decay rates, along with data on human indoor physical activities obtained from the mass balance model, will be incorporated into the CONTAM model for its optimisation [36]. This process will account for variations in housing characteristics, such as building archetypes and ventilation systems. In the final step, we will use the data gathered from the national survey (WP4) to model exposure to PM_2.5_ and NO_2_ within ethnically diverse communities. Furthermore, we will assess the exposure impacts of behaviour changes in domestic fuel use for cooking and heating, smoking/vaping, ventilation settings identified in WP4, as well as the effects of NetZero transitions [37].

A Health Impact Assessment (HIA) will also be conducted under this WP, including a scoping review to identify suitable concentration-response functions between indoor exposures and health endpoints from the literature, starting from expert review documents [38]. In some cases, outdoor air pollution concentration-response functions [39] may be suitable, with some adaptation, to account for known interrelationships between outdoor and indoor concentrations and personal exposure [40]. This health impact assessment will be focussed on the concentrations of pollutants measured within this project, taking averaging times into account, and on pollutants linked to particular sources that are subject to a plausible hypothetical policy scenario. For example, for swapping gas cookers for electric cookers, measurements of NO_2_ from WP2 in 100 households will provide an estimate of the difference in concentration between homes with gas cookers vs. those without. This concentration change can then be linked with appropriate concentration-response functions from the literature, such as from a meta-analysis of associations with children’s respiratory symptoms [41], supported by more recent evidence [42] to develop improved indoor concentration-response functions. The population at risk would be households including children under 12 with a gas cooker for this example. Various sources will be checked for baseline rates including our own study, our previous studies [43,44], the constituent studies within the meta-analysis, the Health Survey for England, and WHO recommendations of concentration-response functions for outdoor air pollution [45]. Similar principles will be applied to other scenarios and pollutants such as increased surveillance for CO levels and reduction in angina symptoms (if a problem is identified); frequency of vacuuming, coarse PM and respiratory endpoints; reducing humidity, endotoxin/bioaerosol measures, and respiratory endpoints, etc. The exact choice of calculations will necessarily be dependent on findings from the study itself, both in terms of identifying the most important sources and pollutants, exposures and any finding of associations with symptoms (WP6) or suggested hazard from toxicological investigations (WP3). The HIA will be applied to the wards relevant to the study area (more frequent health outcomes) and to areas across England with similar population characteristics and housing stock.

### 2.9. Harmonising the Data and Statistical Analysis of Relationships Between Sources Behaviours and Symptoms (WP6)

This WP will use the data collected from the general and health questionnaires, indoor and outdoor pollution sampling, and time-activity and symptom diaries, harmonised in an integrated database. We will estimate children’s personal exposure to multiple pollutants exposure (PM_2.5_, PM_10_, NO_2_, CO, VOCs, formaldehyde), as well as temperature and relative humidity, to estimate their total exposures, while also separating concentrations into indoor- and outdoor-generated pollution following methods described in previous publications. Briefly, we will combine infiltration efficiencies, calculated as the indoor/outdoor ratio of air pollution measurements conducted within WellHome with time-activity and estimate children’s exposures at every 30-min interval. Daily air pollution metrics, such as 24-h mean, 1-h max, 8-h max and others, will be calculated and linked with daily symptoms, medication (inhaler) use, and school absenteeism. Descriptive analysis will be conducted to understand the data structures, including descriptive statistics (e.g., mean/median; standard deviation/interquartile range, maximum, and minimum), distribution plots (e.g., histograms, boxplots), and correlation coefficients.

We will further assess the impact of personal exposure to indoor and ambient temperature and relative humidity on respiratory symptoms, medication use, and absenteeism. Logistic mixed-effects models with a random effect per child will be applied to quantify the associations between multiple exposures (and appropriate lags for the exposures) and health outcomes, adjusting for potential confounders measured in the questionnaires and identified through directed acyclic graphs (DAGs). We will also assess the impact of using different exposure metrics, or proxies such as the measurements from the nearest monitor, on the exposure-response associations.

From the TRACK, PAQLQ, and ACT questionnaires, we have health outcomes related to the previous one to four weeks, repeated in each campaign. We will apply mixed models to investigate the associations of the previous one to four weeks’ averages of personal exposures (including chemicals) with asthma control and quality of life. Similarly, asthma severity and control will be explored in relation to household characteristics and behaviours.

We will further use the integrated WellHome database to develop a statistical model for predicting daily indoor concentrations using the household characteristics and the participants’ behaviours (e.g., cooking/heating type and frequency, window opening, traffic density around their home, etc.) as explanatory variables. Regression analysis and machine learning algorithms will be applied to improve the explained variability of indoor exposures. This model will be combined with the integrated indoor model described above (Section 2.8) using a generalised additive model approach in order to maximise its predictive accuracy, as done before from our group for ambient concentrations.

## 3. Discussion

The WellHome study adopts a transdisciplinary approach in partnership with community members to provide a better understanding of air pollution exposures in children and adults across the indoor-outdoor air pollution continuum, with the ultimate purpose to improve population health by identifying health inequities to reduce hazards for the most vulnerable groups, specifically children with asthma. Gaining a clearer understanding of indoor air pollution in general, and its sources and impacts in particular, will help inform the development of effective measures to reduce exposure and associated risks to vulnerable communities. To achieve this, we will consider not only aerosol mixtures/sources and exposures, but also their socio-economic and behavioural drivers. Establishing key drivers will help inform the co-development of less invasive behavioural interventions that are acceptable and respectful to these communities. This study will then evaluate whether these changes reduce the hazard of breathed in indoor air and whether this results in better respiratory health.

This project therefore adopts novel theoretical and methodological approaches to address knowledge gaps related to indoor air pollution and its impact on asthma. This study will contribute to wider academic research in the field of air quality and mixed methodologies research. Though moving from the outdoor to the indoor environment poses significant scientific challenges, it also presents new opportunities for health improvement. While building high quality homes out of safe materials remains paramount, understanding the factors driving personal and household behaviours affecting residential air quality can amplify these efforts.

The data gathered from this study will enable us to identify specific behavioural interventions that could lead to major improvements. However, these interventions must be acceptable and appropriate to vulnerable and disadvantaged communities, or they risk compounding, rather than alleviating, existing health inequalities. To address this, we will work closely with participants and community members to ensure our approach and recommendations are both feasible and culturally appropriate. By actively involving the community throughout the research process, we can tailor interventions to meet their specific needs and preferences. This collaborative approach not only enhances the relevance and effectiveness of future interventions, but also fosters trust and engagement, paving the way for sustainable improvements in indoor air quality and overall health outcomes.

### Dissemination

In line with our participatory community approach, our findings will be communicated to participants through co-designed activities and reports at the end of the measurement programme. This will be complemented with evidence-based recommendations for reducing air pollution exposure. We will also co-design an online community research hub in collaboration with our local partners and participating households. This hub will provide access to educational materials, case studies, community advice, and air quality data from the Breathe London network. Our objective is to give continuity to this community hub beyond the duration of the study, making it a self-sustaining tool for further air quality research and engagement.

The project’s outputs and progress will be disseminated through peer-reviewed publications, public-facing literature, and presentation at relevant conferences and symposia. We will also organise seminars and workshops within the MRC Centre for Environment & Health and the NIHR Health Protection Research Unit in Environmental Exposures and Health. Additionally, we will run short open courses, which will include a broad overview of our work. We will participate in dissemination activities organised by our funders, the SPF Clean Air Programme and collaborate closely with aligned studies funded under this programme. Anticipated outcomes also include offering clear information to shape effective recommendations for householders, authorities, and policymakers, thus fostering healthier home environments.

## Figures and Tables

**Figure 1 ijerph-22-00249-f001:**
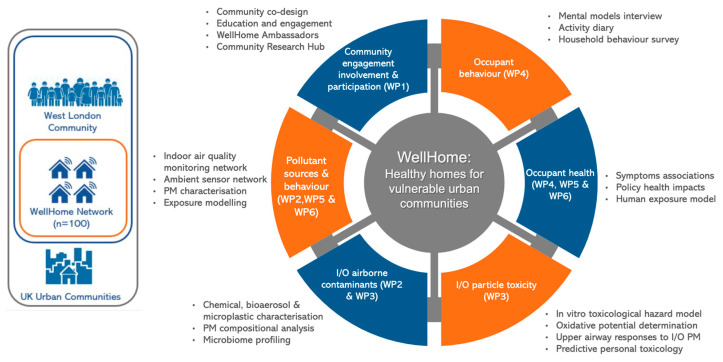
WellHome study schematic illustrating the work packages and associated activities.

**Figure 2 ijerph-22-00249-f002:**
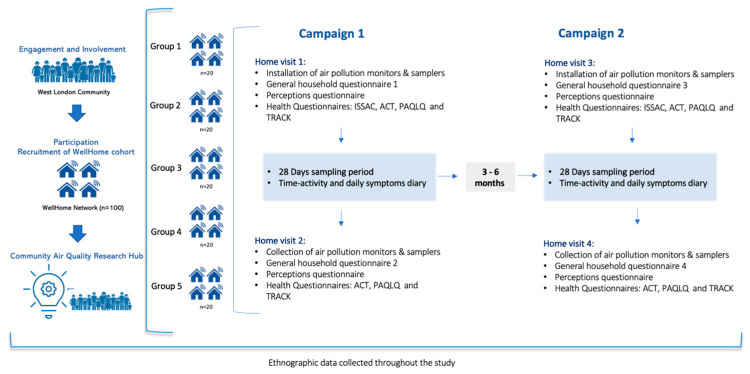
WellHome study design schematic.

**Figure 3 ijerph-22-00249-f003:**
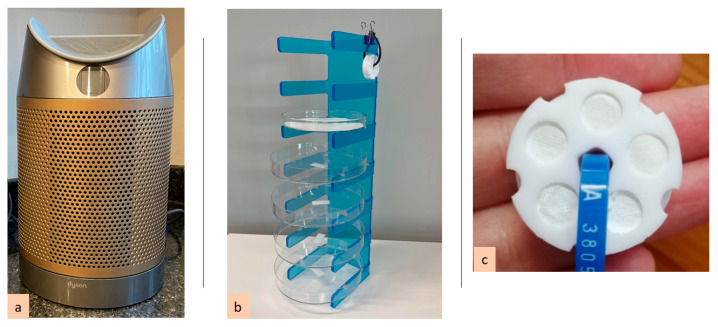
(**a**) Modified Dyson air purifier; (**b**) microplastic and bioaerosol passive deposition samplers; (**c**) chemical passive sampler devices.

## Data Availability

As this study is a protocol, no data are currently available.

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
