# Peer review of "West London Healthy Home and Environment (WellHome) Study: Protocol for a Community-Based Study Investigating Exposures Across the Indoor-Outdoor Air Pollution Continuum in Urban Communities"

_ijerph, 2025, doi:10.3390/ijerph22020249_

Round 1

Reviewer 1 Report

Comments and Suggestions for Authors

 1) For the 100 homes, why were two 28-day periods chosen? When exactly will these periods occur? I understand that not all homes will be monitored simultaneously, but rather split into 5 groups, with 20 homes monitored at a time. Considering the influence of seasonal variations on air pollution, how will you account for potential differences in results among the groups?

2) Regarding the 10 homes with additional 12-month measurements, will 2 be selected per group? What specific criteria will be used for their selection? This needs to be clearly explained to justify and clarify this second part of the project.

3) Lines 180 and 408: Did the ethics committee approve the compensation and the voucher described in the protocol? It is necessary to mention this in the study due to ethical concerns about this context.

4) The project is extensive and multidisciplinary. The protocol description will be refined with details about the team involved and the specific tasks each member will undertake. Will there also be participation from graduate and undergraduate students in research and academic activities?

Author Response

1) For the 100 homes, why were two 28-day periods chosen? When exactly will these periods occur? I understand that not all homes will be monitored simultaneously, but rather split into 5 groups, with 20 homes monitored at a time. Considering the influence of seasonal variations on air pollution, how will you account for potential differences in results among the groups?

Thank you for your question. The goal of the study is to monitor homes during two broad periods across the year: the cool-cold months and the warm-hot months. Each monitoring period spans 28 days, which is equivalent to four weeks or approximately one month of data. These monitoring durations are consistent with other studies and provide a reliable representation of a home’s trends during each general season. For each home and monitoring period, we will collect over 100,000 data points per sensor pollutant.

In terms of seasonal variations in air pollution, data analysis will group the observations into those two general periods for comparison.  While weather patterns in the UK are highly variable within a given season, household behaviour typically remains stable throughout each season. This stability allows for comparisons within a home during a season, as well as broader comparisons between homes across seasons. The study is focused on understanding indoor air quality exposure.

2) Regarding the 10 homes with additional 12-month measurements, will 2 be selected per group? What specific criteria will be used for their selection? This needs to be clearly explained to justify and clarify this second part of the project.

Thank you for your question. The 10 homes (Priority Homes) will be recruited separately from the initial cohort and will have their own inclusion criteria. These criteria include:

  • At least one child in the household must have a diagnosis of asthma by a health practitioner.
  • Participants should expect to reside at the same address for a minimum of 12 months.
  • There must be no smokers in the household.
  • Similar to the 100 homes group, participants must reside in West London.

We have clarified in the manuscript (Page 3- line 126) that the 10 homes group refers to separate households invited to participate in the study, and that they must meet the specific criteria outlined above.

3) Lines 180 and 408: Did the ethics committee approve the compensation and the voucher described in the protocol? It is necessary to mention this in the study due to ethical concerns about this context.

Thank you for your question. We can confirm that our ethics committee has approved the use of participant compensation. We have now included a statement in Page 5 - line 211 indicating that this approach was approved by the ethics committee.

4) The project is extensive and multidisciplinary. The protocol description will be refined with details about the team involved and the specific tasks each member will undertake. Will there also be participation from graduate and undergraduate students in research and academic activities?

Thank you for your observation. We plan to include a number of postgraduate students across the project, across the different work packages. We have now included a statement in Page 6 - line 242 indicating this.

Reviewer 2 Report

Comments and Suggestions for Authors

Manuscript ID: ijerph-3428892. Title: West London Healthy Home and Environment (WellHome) Study: Protocol for a Community-Based Study Investigating Indoor Air Pollution in an Urban Community in London, England.  

Comments:

1. The title of the article is very long. Please synthesize. Additionally, I suggest replacing the city of study by a couple of terms that allow visualizing the novelty of your study.

2. In the abstract the authors should include the main findings of their study. Additionally, the findings in the abstract should be supported with quantitative information from this study. 

3. Please check the following keywords, they could be very general: co-design, housing, and hazard.

4. The objective of this study also mentions outdoor measurements, however, in the title the authors only highlight indoor measurements. This makes the article less coherent. Please check.  

5. The authors should deepen the introduction in relation to the following aspects: what are the pollutants to be prioritized in this study, what is the exposure matrix (respiratory, dermal or ingestion), interaction between pollutants, and passive sampling techniques.  

6. In the introduction, the authors should better highlight the practical usefulness of this study. Possibly, in this work, it is better to have a comprehensive introduction without too many sections.

7. Please synthesize and give more clarity to the objectives of this work. Additionally, Figure 1 should be moved to the materials and methods chapter.

8. Authors should include in section 2.1 a figure showing a map of the study site. This is based on the selection of participants. The context of the study site is relevant to this work.

9. The authors should significantly improve the chapter on materials and methods. References should be included to support all techniques and methods used in this study. For example, what were the standards used to perform the laboratory analyses of the samples collected? Please check and detail this entire chapter. 

10. In section 2.5.1, which standards were considered.

11. Authors should detail the technical aspects of passive sampling equipment, or give a general description and provide a specific reference for a more detailed feature query. Section 2.5.2.

12. In the chapter on materials and methods, the authors should include a figure with a map showing the active and passive sampling sites considered in this work.

13. The authors should significantly improve the chapter on materials and methods. Currently this chapter is very long and is loaded with too many sections that make it difficult to understand. Additionally, the authors should synthesize the titles of some sections.

14. In the chapter on materials and methods, the authors should include technical information to answer the following questions: what descriptive statistical analyses were considered in this study? were multivariate analyses used to study the relationship between variables? what test was used to study the relationship between active and passive monitoring measurements and respondents' perception? was there a spatial analysis in this work? was there a method to validate the questionnaires used in this study? was there a method to validate the results of passive sampling? was there a method to validate the results of passive sampling? All of the above information should be included in this article. 

15. This article does not present a chapter on results. Therefore, the results of this work cannot be analyzed. Apparently, this study is very interesting, but unfortunately the article is incomplete. 

16. The authors should significantly improve the discussion of results. Deepen and contrast their findings with other reference studies. Indeed, this is impossible without a results chapter. They could also include a single chapter on results and discussion.

17. The authors should include a chapter with the main conclusions of this study. In addition, the main limitations detected during the development of this research and future lines of research should be indicated.

18. A manuscript with all of the above comments should be rejected for publication. However, the study is interesting and for this reason I consider that it should be given a second chance to finally evaluate its quality.

Author Response

Thank you for taking the time to review our manuscript and for providing your valuable feedback. We would like to clarify that this manuscript is a protocol. As such, certain concerns, such as the absence of results, additional methodological details, and discussions of findings and their implications, cannot be addressed at this stage. However, we greatly appreciate your insights and will ensure they are considered in follow-up publications where the study results are presented. Your comments will help us improve the quality and scope of our future work.

  1. The title of the article is very long. Please synthesize. Additionally, I suggest replacing the city of study by a couple of terms that allow visualizing the novelty of your study.

Thank you for your suggestion. We have reviewed and revised our study title:

“West London Healthy Home and Environment (WellHome) Study: Protocol for a community-based study investigating exposures across the indoor-outdoor air pollution continuum in urban communities”.

  1. In the abstract the authors should include the main findings of their study. Additionally, the findings in the abstract should be supported with quantitative information from this study. 

Thank you for your feedback. However, as noted above, this manuscript is a protocol rather than the results of a completed study. As such, findings cannot be included in the abstract, and quantitative information is not yet available. The abstract instead focuses on the aims, design, and planned methods of the study, consistent with the purpose of a protocol.

  1. Please check the following keywords, they could be very general: co-design, housing, and hazard.

Thank you for your Highlighting this, we have reviewed and revised our keywords.

indoor air pollution; exposure modelling; asthma; respiratory symptoms; behaviour change; immunotoxicity; participatory research.

  1. The objective of this study also mentions outdoor measurements, however, in the title the authors only highlight indoor measurements. This makes the article less coherent. Please check.

Thank you for your suggestion. We have reviewed our study title:

“West London Healthy Home and Environment (WellHome) Study: Protocol for a community-based study investigating exposures across the indoor-outdoor air pollution continuum in urban communities”.

  1. The authors should deepen the introduction in relation to the following aspects: what are the pollutants to be prioritized in this study, what is the exposure matrix (respiratory, dermal or ingestion), interaction between pollutants, and passive sampling techniques.

Thank you for your comment. Additional details have now been included throughout the introduction and the materials and methods section (Section 2).

  1. In the introduction, the authors should better highlight the practical usefulness of this study. Possibly, in this work, it is better to have a comprehensive introduction without too many sections.

Thank you but we feel that we have adequately covered the purpose of the study – to comprehensively measure and understand indoor air quality sources and their impacts on respiratory function of asthmatic children. The practical significance of this research—specifically its potential to inform the development of effective measures and interventions to reduce exposure and associated risks for vulnerable communities—has been emphasised in the discussion section. We trust this sufficiently captures both the scope and the application of our findings.

  1. Please synthesize and give more clarity to the objectives of this work. Additionally, Figure 1 should be moved to the materials and methods chapter.

Thank you. The study's objectives, which will be collectively addressed across its six complementary work packages, are explicitly listed in the aptly named “Study Aim and Objectives” section (1.3) on pages 3–4. We have reviewed your suggestion and believe that Figure 1 should remain in its current section, as it is closely linked to the aim and objectives. It effectively illustrates how the work packages complement each other to address these objectives.

  1. Authors should include in section 2.1 a figure showing a map of the study site. This is based on the selection of participants. The context of the study site is relevant to this work.

Thank you for this suggestion. However, as participant selection and recruitment have not yet taken place, we are unable to provide a map of the study site at this stage. This information, including a detailed map, will be included in future publications.

  1. The authors should significantly improve the chapter on materials and methods. References should be included to support all techniques and methods used in this study. For example, what were the standards used to perform the laboratory analyses of the samples collected? Please check and detail this entire chapter.

Thank you for your suggestion we have now provided further details regarding our methods in pages 7 and 8 - section 2.5.2.

  1. In section 2.5.1, which standards were considered.

Thank you for your observation. Measurements will be compared to established indoor air quality standards (e.g. BS40102). We have incorporated this information in page 7  line 301.

  1. Authors should detail the technical aspects of passive sampling equipment, or give a general description and provide a specific reference for a more detailed feature query. Section 2.5.2.

Thank you for your suggestion. We have now added additional details in pages 7 and 8 - section 2.5.2.

  1. In the chapter on materials and methods, the authors should include a figure with a map showing the active and passive sampling sites considered in this work.

Thank you for your suggestion. As this manuscript outlines a protocol, the active and passive sampling sites have not yet been finalised. We will include a detailed map of these sites in future publications once the study is conducted and data collection is complete.

  1. The authors should significantly improve the chapter on materials and methods. Currently this chapter is very long and is loaded with too many sections that make it difficult to understand. Additionally, the authors should synthesize the titles of some sections.

Thank you for your feedback. We believe that presenting the materials and methods in their current format is the best way to highlight the multiple components and work packages that collectively address the research question. As this is a protocol, the goal is to provide an overview of the planned activities and methods. Future publications will focus on specific aspects of the study, allowing for more detailed discussions and deeper exploration of individual activities and methods.

  1. In the chapter on materials and methods, the authors should include technical information to answer the following questions: what descriptive statistical analyses were considered in this study? were multivariate analyses used to study the relationship between variables? what test was used to study the relationship between active and passive monitoring measurements and respondents' perception? was there a spatial analysis in this work? was there a method to validate the questionnaires used in this study? was there a method to validate the results of passive sampling? was there a method to validate the results of passive sampling? All of the above information should be included in this article. 

Thank you for your feedback. You are correct, and all of this information will be included in follow-up publications. However, as this manuscript outlines a study protocol, these aspects are not applicable at this stage. The protocol is intended to provide an overview of the planned study design, while detailed analyses, validations, and specific methods will be addressed in future publications once the study has been conducted.  At this stage, however, we can provide additional details regarding our descriptive analysis, which we have included in Page 13 – line 574.

  1. This article does not present a chapter on results. Therefore, the results of this work cannot be analyzed. Apparently, this study is very interesting, but unfortunately the article is incomplete. 

Thank you for taking the time to review our manuscript. As per before, we would like to clarify that this manuscript is a protocol, and therefore, we do not have results at this stage.

  1. The authors should significantly improve the discussion of results. Deepen and contrast their findings with other reference studies. Indeed, this is impossible without a results chapter. They could also include a single chapter on results and discussion.

Thank you for taking the time to review our manuscript. As mentioned previously, we would like to clarify that this is a protocol manuscript, and as such, results are not available at this stage.

  1. The authors should include a chapter with the main conclusions of this study. In addition, the main limitations detected during the development of this research and future lines of research should be indicated.

Thank you for taking the time to review our manuscript. As mentioned previously, we would like to clarify that this is a protocol manuscript. We will ensure that follow-up publications address limitations, lessons learned, and potential areas for future research.

  1. A manuscript with all of the above comments should be rejected for publication. However, the study is interesting and for this reason I consider that it should be given a second chance to finally evaluate its quality.

We sincerely appreciate the time and effort you dedicated to reviewing our protocol. We regret that it may not have been entirely clear that this is a protocol paper, and we hope the journal might consider emphasising this more explicitly to reviewers in the future to streamline and enhance the review process.

We look forward to sharing the insights from our work in future publications and hope you will have the opportunity to read them.

Reviewer 3 Report

Comments and Suggestions for Authors

Title: West London Healthy Home and Environment (WellHome) Study: Protocol for a Community-Based Study Investigating Indoor Air Pollution in an Urban Community in London, England.

The study from Varaden et al. proposes a protocol to investigate indoor air pollution in an urban community in London, England. The interest in the subject stems from the need to improve air quality in kitchens, children's bedrooms, and living rooms through a better characterization of multiple physical, chemical, microplastic, and biological contaminants. To my knowledge, this is the first field study focusing on the improvement in air quality indoors. I find the protocol outcome of great value and interest to the atmospheric chemistry research and health community and in scope with the journal. I recommend the protocol be published and only have some minor comments to make some points clearer to the readers.

Lines 313-315. The authors declare the study aims to measure VOCs and semi-VOCs using non-target screening gas chromatography-mass spectrometry.

Lines 464- 465. The authors claim to use a bass balance model for a dataset of air pollutants (PM2.5, PM10, and NO2).

The two paragraphs raise one question. Do the experiments measure the abundant components (e.g., mass concentration, ions, metals, and carbonaceous aerosols) of air pollutants (PM2.5 and PM10)?

Author Response

The study from Varaden et al. proposes a protocol to investigate indoor air pollution in an urban community in London, England. The interest in the subject stems from the need to improve air quality in kitchens, children's bedrooms, and living rooms through a better characterization of multiple physical, chemical, microplastic, and biological contaminants. To my knowledge, this is the first field study focusing on the improvement in air quality indoors. I find the protocol outcome of great value and interest to the atmospheric chemistry research and health community and in scope with the journal. I recommend the protocol be published and only have some minor comments to make some points clearer to the readers.

Lines 313-315. The authors declare the study aims to measure VOCs and semi-VOCs using non-target screening gas chromatography-mass spectrometry. 

Lines 464- 465. The authors claim to use a bass balance model for a dataset of air pollutants (PM2.5, PM10, and NO2). 

The two paragraphs raise one question. Do the experiments measure the abundant components (e.g., mass concentration, ions, metals, and carbonaceous aerosols) of air pollutants (PM2.5 and PM10)?

Thank you for reviewing our protocol and for your positive feedback. We have chemical profiles of outdoor PM2.5 from a nearby monitoring station. However, we will only collect high-time resolution indoor measurements of mass concentrations (PM2.5 and PM10) and the number size distribution of particulate matter (PM) for the 10 priority homes. We will apply a mass balance model using high-time resolution data to estimate infiltration factors and emission rates of PM mass. We will ensure further details about this are presented in follow up publications.

Regarding the measurement of the VOCs and sVOCs, this study aims to capture the chemical diversity of the indoor and outdoor air environments by qualitatively measuring ion abundances of volatile chemicals recovered from the passive sampler devices in each home and outdoors.  We have now added further details about this in Pages 7 and 8 Section 2.5.2.

Additionally, insights into the abundant components of particulate matter (PM2.5 and PM10) may be uncovered by the experiment involving pyrolysis-GC-MS assessment of microplastic concentrations in the air. This is due to the potential of the pyrolysis process to break down PM (deposited on the filter) into its constituent chemical components.

We will ensure that these specific details are included in future publications once we have completed our analysis.

Round 2

Reviewer 2 Report

Comments and Suggestions for Authors

Manuscript ID: ijerph-3428892-R2. Title: West London Healthy Home and Environment (WellHome) Study: Protocol for a Community-Based Study Investigating Indoor Air Pollution in an Urban Community in London, England.  

Comments:

In this new version of the article, the authors did not adequately address each of the 18 comments made in the previous round of revisions. Unfortunately, the article does not present the technical details of the protocol used and the absence of results makes the article of no interest to the journal's audience. Therefore, I suggest rejecting the article for publication.

Kind regards.